# Biogas Production from Microalgal Biomass Produced in the Tertiary Treatment of Urban Wastewater: Assessment of Seasonal Variations

**Raúl Barros** [1,*], **Sara Raposo** [1], **Etiele G. Morais** [2], **Brígida Rodrigues** [1], **Valdemira Afonso** [1], **Pedro Gonçalves** [1], **José Marques** [2], **Paulo Ricardo Cerqueira** [2], **João Varela** [2,3], **Margarida Ribau Teixeira** [4] **and Luísa Barreira** [2,3,*]

1    CIMA—Centre for Marine and Environmental Research, FCT, Campus de Gambelas, Universidade do Algarve, 8005-139 Faro, Portugal
2    CCMAR—Centre of Marine Sciences, Campus de Gambelas, Universidade do Algarve, 8005-139 Faro, Portugal
3    GreenCoLab—Green Ocean Technologies and Products Collaborative Laboratory, CCMAR, Universidade do Algarve, 8005-139 Faro, Portugal
4    CENSE—Centre for Research on the Environment and Sustainability & CHANGE—Global Change and Sustainability Institute, Campus de Gambelas, Universidade do Algarve, 8005-139 Faro, Portugal
*    Correspondence: rbarros@ualg.pt (R.B.); lbarreir@ualg.pt (L.B.)

**Abstract:** The valorization of microalgal biomass produced during wastewater treatment has the potential to mitigate treatment costs. As contaminated biomass (e.g., with pharmaceuticals, toxic metals, etc.) is often generated, biogas production is considered an effective valorization option. The biomass was obtained from a pilot facility of photobioreactors for tertiary wastewater treatment. The pilots were run for one year with naturally formed microalgal consortia. The biogas was generated in 70 mL crimp-top vials at 35 °C, quantified with a manometer and the methane yield measured by gas chromatography. A maximum biogas production of 311 mL/g volatile solids (VS) with a methane yield of 252 mL/g VS was obtained with the spring samples. These rather low values were not improved using previous thermo-acidic hydrolysis, suggesting that the low intrinsic biodegradable organic matter content of the consortia might be the cause for low yield. Considering the total volume of wastewater treated by this plant and the average amount of methane produced in this study, the substitution of the current tertiary treatment with the one here proposed would reduce the energy consumption of the plant by 20% and create an energy surplus of 2.8%. The implementation of this system would therefore contribute towards meeting the ambitious decarbonization targets established by the EU.

**Keywords:** biomethane; thermo-acidic hydrolysis; microalgal composition; energy potential

## 1. Introduction

Microalgae are considered an attractive source of biomass to produce biofuels and high-value products. They have the advantages of fast and high reproduction rates and sustainable growth, and no competition with food for arable land [1]. Depending on the media composition, the microalgae biomass can be rich in carbohydrates (30–60%), lipids (10–30%) and proteins (20–70%) [2]. Biofuel production by microalgae is still an underdeveloped area all over the world. One of the main drawbacks is related to biomass production costs. The use of alternative sources of nutrients, recycling of effluents and use of biorefinery systems are alternatives to reduce the production costs of microalgae biofuels [3,4].

The coupling of wastewater treatment and microalgae cultivation is gaining attention, as it promises to be an environmentally friendly pathway to recover nutrients with microalgae biomass as a co-product of the process [5]. Conventional wastewater (WW) treatment

systems are expensive, use a lot of energy, emit large amounts of greenhouse gases and do not recover nutrients such as nitrogen or phosphorus present in the wastewater. The use of microalgae for treatment is a promising alternative as they are able to capture the nutrients from the WW for cell development and multiplication, opening possibilities for their recycling. Nitrogen, carbon, and phosphorus are the main components present in WW and are also those that represent the highest cost in microalgae production. Furthermore, WW treatment using microalgae emits less greenhouse gas than conventional treatments [6]. In contrast to conventional nitrification/denitrification technologies, the use of microalgae for tertiary wastewater treatment allows the capture of nitrogen, with the potential for its reuse in agriculture or other activities, reducing the need for synthetic fertilizers.

The main components still present in urban wastewater after secondary treatment are nitrogen and very low amounts of organic carbon and phosphorus. For this reason, microalgae consortia used as a tertiary wastewater treatment system produce low lipid amounts but can be rich in proteins and carbohydrates [5]. In these systems, anaerobic digestion (AD) appears to be the most promising method to recover energy from biomass, which is an appropriate renewable substrate to generate biogas. The production of biogas from microalgae is an attractive area, as it can generate up to 1 kWh of electricity/kg VS [7,8]; it does not require the extraction of specific macromolecules (lipids, proteins, or carbohydrates) and can be performed using wet biomass [9].

AD is a complex process involving the different sequential metabolic stages of hydrolysis, acidogenesis, acetogenesis, and methanogenesis [10–12], which influence biogas yield. In this process, the organic matter is decomposed in the absence of oxygen ($O_2$) by a microbial consortium of different species with competence to carry out all those metabolic activities, producing biogas and digestate as a by-product [10,13]. It is also worth mentioning that microalgae can be used at biogas sites to clean up biogas or gas emissions by carrying out $CO_2$ sequestration, or to recover nutrients from liquid digestates [14–17].

Some microalgae have low anaerobic biodegradability because of their complex cell structure, which determines their accessibility to organic matter and, hence, the rate of the AD hydrolysis. To enhance biogas yield, sometimes it is necessary to apply a pretreatment step to facilitate cell disruption, increase the reaction rate, and enhance methane yield [18]. Thermal pretreatment operations have been shown to efficiently disrupt algal cells. This process is performed at low temperatures (75–95 °C) and was successfully used to enhance microalgae anaerobic biodegradability [13]. However, the best efficiencies of intracellular bioproduct release were achieved using dilute acid methods [19–22] that are faster and cheaper, even though they have the disadvantage of increasing the salt content of the material [19,21,22]. The enzymatic methods are usually slower and more expensive, [18,23]. The combination of a thermal method with the action of a weak acid has been widely investigated as a sustainable pretreatment technology appropriate to the specificities of different microalgae species.

While most published work has been conducted in single-species systems, tertiary WW treatment with microalgae generally involves a consortium of different species, making it more difficult to predict biogas production yields or even if biogas production is feasible. In this context, the objective of this work was to evaluate biogas production using the biomass of microalgae consortia produced as co-products of a pilot tertiary wastewater treatment system installed inline on a wastewater treatment plant using the novel GreenDune photobioreactors. The study was carried out by allowing the spontaneous blooming of microalgal biomass in different seasons of the year. Biogas production yields, and composition, along with total energetic potential of the system were estimated.

## 2. Materials and Methods

### 2.1. Biomass Harvest from GreenDune Wastewater Treatment System

The biomass was a co-product of the tertiary wastewater treatment produced during a year in a pilot facility of GreenDune photobioreactors (Bluemater, Ecoefficient solutions, Porto, Portugal) installed inline at the Quinta do Lago wastewater treatment plant located

in the South of Portugal (Algarve). The GreenDune photobioreactors were fed with water effluent from the secondary settler, with the objective of replacing an aerated nitrification bioreactor and a methanol-fed Biofor denitrification System. The GreenDune are innovative prismatic open photobioreactors (Figure 1) with a volume of 480 L occupying an area of 1 m². For this experiment, 3 reactors were interconnected providing a total volume of 1440 L. The WW treatment experiments were conducted in 2020 in the different seasons with a variation in the hydraulic retention time (HRT) for winter (24 and 48 h) and a fixed HRT of 24 h for summer, autumn and spring. The wastewater composition over a year varies from: total nitrogen: 20.2–61.3 mg/L; nitrates: 2.7–90.7 mg/L; ammonium: 0.5–30.6 mg/L; total phosphorus: 1.2–5.0 mg/L and chemical oxygen demand: 28.9–150.3 mg/L. Wastewater treatment was performed by a spontaneous microalgal consortium that developed under different HRTs as tested in different seasons of the year. Cultures were conducted in continuous mode for 12 days after a stabilization period of 7 days.

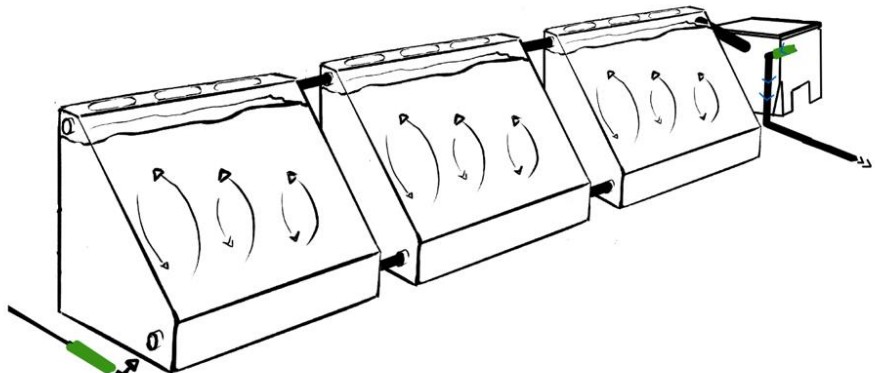

**Figure 1.** A line setup of 3 sequential GreenDune photobioreactors (Bluemater, Ecoeficient Solutions, Porto, Portugal.) totalling 1440 L, discharging into a biomass settler.

The photobioreactor system was equipped with a 50 L settler to collect biomass and a buffer tank for the separation of the treated water. Five liters of the settled biomass was collected every 2 days centrifuged for 10 min at $12,000 \times g$ at 20 °C and pooled giving a sample representative of each experiment (season or HRT). Microalgal biomass dry weight was determined following the APHA method 2540 A for total solids (TS). Lipid content was determined by the gravimetric method of Bligh and Dyer [24] after extraction with a mixture of chloroform, methanol, and water (2:2:1), under homogenization with an IKA Ultra-Turrax disperser (IKA-Werke GmbH, Staufem, Germany) in an ice bath. Protein content was determined by CHN elemental analysis, according to the procedure provided by the manufacturer using Vario EL III equipment (Vario EL, Elemental Analyzer system, GmbH, Hanau, Germany). The final protein content was calculated by multiplying the percentage of nitrogen by a factor of 4.87 [25]. The ash content was determined by burning the freeze-dried biomass in a furnace (J. P. Selecta, Sel horn R9-L, Barcelona, Spain) at 525 °C for 12 h [26]. The carbohydrate content was determined by subtracting ashes, lipids, and proteins content from 100%.

### 2.2. Anaerobic Digestion of Microalgal Biomass

All anaerobic digestion (AD) assays were performed in crimp-top digestion vials, with a working volume of 70 mL and headspace of 52.6 mL. The inoculum used for biogas production was obtained from the anaerobic digestion reactor of Lagos wastewater treatment plant, operated by Águas do Algarve, SA., and the consortia biomass samples were used as substrate. Biomass of *Skeletonema costatum* produced by Necton (Companhia Portuguesa de Culturas Marinhas s.a.) [27] was used as a control for biogas production for comparison between a consortium and a monoculture. The inoculum and biomass concentrations were kept constant in all the conditions and assays, at 9 g TS/L and 10 g TS/L, respectively. The substrate-to-inoculum ratio (in VS) varied between 1.07 (for the autumn sample, the

one with the highest ash content) and 1.27 for the summer sample (with the lowest ash content). The final volumes were adjusted with deionized water. Inoculum control blanks were prepared using deionized water instead of biomass to determine the endogenous biogas production of the inoculum. To ensure anaerobic conditions inside the vials, the headspace was purged with nitrogen for 1 min prior to digestion. The digestion vials were incubated in an orbital shaking incubator (INCU-Line; ILS 6; VWR, Lisbon, Portugal), at 120 rpm and 35 °C.

### 2.2.1. Biogas Production, Monitoring, and Energetic Capacity Evaluation

Biogas accumulation was monitored daily by registering the pressure in the headspace of the vials with a manometer (FB Traceable Manometer 0-30PSI; Fisher Scientific; Lisbon, Portugal). To prevent any losses, the biogas was collected into gas-sampling bags every time it reached 300 mBar pressure inside the vial. Biogas accumulation was calculated using Equation (1),

$$V_{biogas} = V_{headspace} \cdot ((P_2 - P_1) \cdot 273.15)/(1000\ T_w) \tag{1}$$

where $V_{biogas}$ is the volume, in mL, of gas produced between measurements under standard temperature and pressure conditions (0 °C and 1 bar), $P_1$ and $P_2$ are the initial and final pressure measurements in mbar, and $T_w$ is the working temperature, in Kelvin. The results reported are computed by discounting the endogenous inoculum biogas production of the blanks and expressing the result per g volatile solids.

### 2.2.2. Thermoacid Hydrolysis Pretreatment

Thermoacid hydrolysis was performed by suspending 50 g/L of biomass samples in 2% (*v/v*) HCl, in triplicate, followed by autoclaving the suspensions for 30 min at 120 ± 1 °C. Samples collected before and after hydrolysis were neutralized with NaOH for further analysis.

### 2.2.3. Reducing Sugars Quantification

Total reducing sugars (TRS) were quantified, before and after hydrolysis, by following the dinitrosalicylic acid (DNS) colorimetric method described by Miller et al. [28], with glucose as standard. The total reducing sugars content of the biomass was calculated by Equation (2) as follows:

$$TRS\ (\%) = (Reducing\ sugar\ (g/L))/(Dried\ Biomass\ (g/L)) \times 100 \tag{2}$$

### 2.2.4. Methane Quantification

Methane was quantified by gas chromatography (Trace 1300, Thermo Scientific, Lisbon, Portugal), using a capillary column (TG-BOND Msieve 5Å, 30 m × 0.53 mm × 50.0 μm, Thermo Scientific, Lisbon, Portugal) and a thermal conductivity detector (TCD). Biogas was injected into the system using a gas syringe. Helium was used as carrier gas with a flow rate of 3 mL/min. The working temperatures were defined at 40 °C in the oven and column, 76 °C in the injector and 200 °C in the detector, with an increment of 15 °C/min. All the measurements were made in triplicate, and the data was analyzed with NAS UniChrome V software (New Analytical Systems ltd, Minsk, Belarus).

### 2.2.5. Chemical Oxygen Demand (COD)

Chemical oxygen demand of the digestion mixture was determined before and after anaerobic digestion, by following the method described in APHA 5220 D.

### 2.2.6. Statistical Analysis

All experiments were carried out in triplicate and the data were expressed as the mean ± standard deviation. GraphPad Prism version 9.2.0 (GraphPad Software, Inc., San Diego, CA, USA) was used for all statistical analysis. One-way analysis of variance

(ANOVA) followed by Tukey's multiple comparison test $p < 0.05$ was used to examine the differences between the individual conditions of the first line of Table 2 (biogas production from samples of different seasons). Two-way analysis of variance (ANOVA) followed by Tukey's multiple comparison test $p < 0.05$ was used to examine the difference among individual conditions of the second and third lines of Table 2 (methane production from samples of different seasons with or without acid hydrolysis).

## 3. Results and Discussion

### 3.1. Biomass Composition

Table 1 shows the biochemical composition of the biomass subjected to anaerobic digestion collected from the different experiments performed in the different seasons and with two different HRTs in the winter. This table also shows the composition of the biomass of a *Skeletonema* sp. monoculture grown under standardized conditions. Concerning the samples collected from the GreenDune system, the main differences were found in the spring and summer samples, which had higher protein (37 and 40%, respectively, compared to 25–26% for the other samples) and lipid (12 and 8%, respectively, compared to 5–7% for the other samples) contents. Conversely, the ash content of the summer sample was lower (20% vs. 27 to 33% for the other samples). This variability reflects the variations of the composition of the influent WW as well as the temperature and solar radiation differences observed throughout the seasons. The operational conditions seemed to have little effect on biomass composition, since the winter samples operated at HRTs of 24 or 48 h had a very similar biochemical composition. As a reference for comparison, *Skeletonema* biomass, grown under optimum growth conditions, shows a lower protein content (22%) and high lipid (14%) and ash (32%) contents. This probably results from a culture condition with abundant carbon availability, while the relatively high ash content reflects the presence of a silicified frustule, typical of diatoms [29].

**Table 1.** Biomass composition (% *w/w*).

|  | Summer | Autumn | Spring | Winter24 | Winter48 | *Skeletonema* * |
|---|---|---|---|---|---|---|
| Proteins | $39.5 \pm 0.8$ | $24.9 \pm 0.4$ | $36.7 \pm 1.9$ | $26.1 \pm 0.9$ | $25.2 \pm 0.3$ | $22.2 \pm 3.7$ |
| Carbohydrates | $29.3 \pm 0.5$ | $35.5 \pm 2.0$ | $25.3 \pm 0.9$ | $41.8 \pm 0.9$ | $39.4 \pm 2.1$ | $29.6 \pm 2.8$ |
| Lipids | $11.5 \pm 0.3$ | $6.7 \pm 1.3$ | $7.7 \pm 0.5$ | $5.4 \pm 0.2$ | $6.2 \pm 1.6$ | $14.1 \pm 1.9$ |
| Ash | $19.8 \pm 0.4$ | $32.9 \pm 1.4$ | $30.3 \pm 0.2$ | $26.7 \pm 0.9$ | $29.2 \pm 0.9$ | $32.1 \pm 2.4$ |

* Data from Maia et al. [27].

### 3.2. Biogas Production

The time evolution of biogas production for the different biomass samples subjected to anaerobic digestion is shown in Figure 2. For all conditions, an increase in the biogas production rate was observed during the first 15 days, as the microbial consortium on the inoculum better adapts to the biomass being digested. After this period, there was a decrease in biogas production rate as the most easily biodegradable organic matter was gradually depleted. The low biogas production rates observed after day 45–50 indicate that the anaerobic digestion was essentially complete after this time (Figure 2). The methane content of the biogas produced ranged between 60 and 70% (*v/v*) for the digestion of biomass samples but was only 43% (*v/v*) for the inoculum control.

Comparing the biogas production for the different seasons with the same hydraulic retention time (24 h) (Figure 2A and Table 2), the spring sample showed the highest production (311 mL/g of volatile solids) with no significant difference from summer (258 mL/g of volatile solids). The lowest biogas production obtained was for the autumn sample (172 mL/g of volatile solids) and winter sample (167 mL/g of volatile solids). The highest methane production was also apparently observed with the spring sample (Table 2); however, there were no statistically significant differences between the methane production of the anaerobic digestions carried out with the different samples of non-hydrolyzed biomass. Vargas-Estrada et al. [30] investigated the biogas yield of biomass produced as a wastewater treatment co-product and reached maximum production of 204 mL/g volatile

solids, which is within the range observed in this study. The higher yields in summer and spring can be explained by the higher protein (40 and 37%, respectively) and lipid (12 and 8%, respectively) contents in biomass for these seasons compared to the others (Table 1). According to Sialve et al. [31], theoretical methane yields are higher for lipids (1.014 L/g of volatile solids) and proteins (0.851 L/g of volatile solids) than for carbohydrates (0.415 L/g of volatile solids). In practise, the methane yield from microalgae biomass can vary widely depending on the species or consortium composition, medium composition and growth conditions. The reported results range from 0.09 to 0.45 L $CH_4$/g of volatile solids [31].

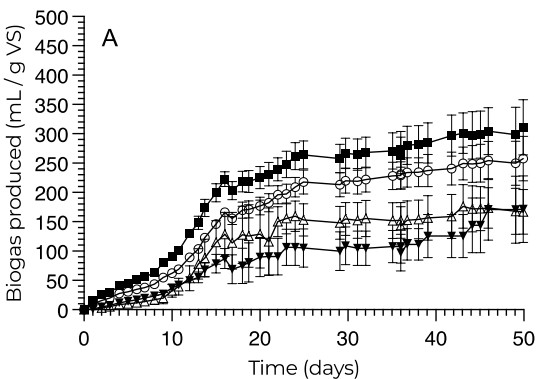 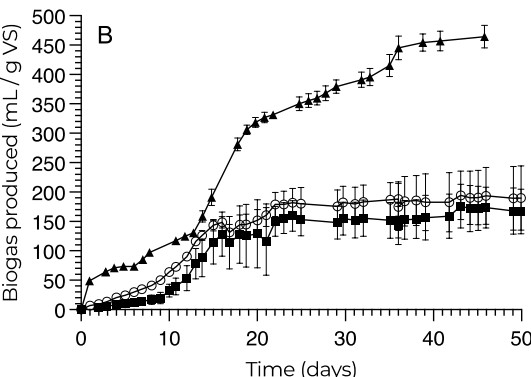

**Figure 2.** Accumulated biogas production after 50 d: (**A**) different seasons (○) summer (▼) autumn (■) spring (△) winter and (**B**) winter season with different HRT (■) 24 h and (○) 48 h and (▲) *Skeletonema* monoculture. The data plotted are the differences between the means of the digested sample and the mean of the inoculum control (all *n* = 3). Error bars represent standard deviation.

**Table 2.** Maximum biogas and methane production (mL/g volatile solids) using non-hydrolyzed (NH) and hydrolyzed (H) biomass.

| Product | Summer | Autumn | Spring | Winter24 | Winter48 | *Skeletonema* |
|---|---|---|---|---|---|---|
| Biogas NH | 258 ± 38 [b,c] | 172 ± 57 [c] | 311 ± 47 [b] | 167 ± 37 [c] | 190 ± 55 [b,c] | 464 ± 19 [a] |
| Methane NH | 211 ± 97 [*,#] | 155 ± 32 [*,#] | 252 ± 36 [*,#] | 149 ± 24 [*,#] | 135 ± 63 [*,#] | - |
| Methane H | 189 ± 8 [#] | 81 ± 11 [#] | 239 ± 12 [#] | 191 ± 29 [#] | 137 ± 21 [#] | - |

The reported results are the difference between the means of the digested sample and the mean of the inoculum control (all *n* = 3) ± the respective standard deviation. The same letters ([a,b,c] used to compare biogas production), or symbols (* used to compare methane production between all non-hydrolyzed biomass conditions; # used to compare methane production between non-hydrolyzed and hydrolyzed conditions of the same season) indicate no statistical difference between tested conditions.

The different hydraulic retention times studied in winter (48 h and 24 h), had no influence on the biogas production potential from the recovered biomass (Figure 2B, Table 2). The composition of these biomass samples did not show any significant differences (Table 2), which helps to explain the similar biogas production. The biomass sample composition was more affected by seasonal changes than by HRT, as the cultures were carried out under outdoor conditions and with wastewater composition and environmental parameters such as temperature and radiation varying during the year, and this is directly related to biogas production (Figure 2). Comparing the production curves from the biomass produced in wastewater treatment with biomass from a monoculture submitted to the same anaerobic digestion process, the monoculture has resulted in a significantly higher amount of biogas reaching 464 mL/g volatile solids (Figure 2B and Table 2).

This is expected because the GreenDune biomass samples grow under suboptimal conditions subjected to competition from different microalgal species but also to grazing due to the presence of predatory or parasitic (micro)organisms. These effects leave a lower proportion of biodegradable organic matter on the volatile solids of the biomass which, after predation or parasitism, consists of material richer in recalcitrant organics, with a lower availability of biodegradable organic matter for anaerobic digestion [32].

To find out if the lower production of biogas was due to the protection provided by the cell envelope to the biodegradable organic matter present within the cells and on the cell walls, thermoacidic pretreatment was applied to the biomass samples, in an attempt to disrupt the cell envelope. This is a rather general method which is able to partially mobilize the organic matter content of microalgae cells for further biodegradation by weakening the cell envelope [19,23,33]. In all samples tested, the reducing sugar content of the pretreated material increased at least four-fold compared to samples without pretreatment. Table 2 shows the results of methane production obtained by anaerobic digestion of the biomass samples, both unprocessed (NH) and subjected to thermoacidic hydrolysis (H).

No significant differences in methane production were observed between the unprocessed (NH) and hydrolyzed (H) biomass samples. In no case did the thermoacidic hydrolysis significantly increase the biomethane production of the biomass samples. These results show that the low production of biogas from the biomass samples collected from the pilot photobioreactor systems is probably not due to the protection of biodegradable matter within the cell envelope of the grown biomass. In fact, the digestion inoculum used seemed to be efficient at hydrolyzing the biomass samples and mobilizing this complex biomolecule matrix for anaerobic digestion, as the unprocessed samples produced approximately the same amount of methane as the biomass samples subjected to thermoacidic hydrolysis. Other authors obtained similar results; as described by Vargas-Estrada et al. (2021), pretreatments are not always efficient in increasing methane yield [34].

### 3.3. Energy Potential of the Generated Biomass

The wastewater treatment plant where the GreenDune pilot system was installed has treated an average 1.36 million $m^3$ of wastewater per year in 2019 and 2020, with a specific power consumption of 1.03 kWh/$m^3$, out of which 0.20 kWh/$m^3$ are used in the tertiary treatment (data provided by the managing company, Águas do Algarve, S.A., Faro, Portugal). Our estimates indicate that if all this wastewater were subjected to tertiary treatment in a full scale GreenDune system, a total of 50 metric tons of biomass would be produced. This is a rather low amount, but it results from tertiary (nutrient removal) treatment only. Considering the averaged production of methane by anaerobic digestion of 130 mL (STP)/g biomass, obtained in the present study, and the lower heating value of 50 MJ/kg for this gas [29], the energy content of the estimated biogas production would be 31.8 MWh per year. The adoption of this novel tertiary treatment technology would reduce the power consumption of the wastewater treatment plant by 20%, while producing an energy surplus that would be equivalent to 2.8% of the total power consumption of the other plant operations. Furthermore, this system precludes the use of methanol for denitrification. This surplus energy is an apparent feature of microalgae-based wastewater treatment systems, as has been demonstrated in the All-Gas project [35]. In that project, an estimated energy return on investment (ratio between energy produced and energy used by the system) of 2.0 was estimated. These systems also have, as an advantage, the sequestration of carbon via photosynthesis.

This is a quite different scenario from the conventional paradigm of wastewater treatment, which uses up a lot of energy, mainly as electric power, and emits large amounts of greenhouse gases. This change in performance justifies by itself that a further effort in research, development and innovation effort is applied to the use of microalgae-based systems for wastewater treatment to help society to comply with the ambitious decarbonization goals required to mitigate human-caused global climate change.

### 4. Conclusions

The main conclusions of this study concerning the anaerobic digestion of microalgal biomass collected from this tertiary pilot treatment system are:

- The biogas production potential is only marginally dependent on seasonality: the biomass produced in the novel pilot GreenDune photobioreactor system applied to tertiary wastewater treatment had a maximum biogas production of 311 mL/g VS with

a methane yield of 252 mL/g VS with the spring samples. No significant difference was observed from the production with the summer samples (258 mL/g VS).

- The biogas production potential is not dependent of the hydraulic retention time under which the system is operated as the composition of the biomass samples did not show any significant differences.
- This biomass has a biogas production potential lower than that of purposefully cultivated microalgal species.
- This low biogas production potential is probably not related to difficulties in mobilizing biodegradable organic matter by the anaerobic digestion inoculum, but rather reflects a low intrinsic biodegradability of the volatile solids contained in the biomass, given the type of microalgae consortia formed.

Nonetheless, this work shows that this process would generate an energy surplus of 2.8% of the total power consumption of the wastewater treatment from the tertiary treatment only, carried out in photobioreactors with negligible energy use. Additionally, these systems do not require the use of methanol and have, as an advantage, the sequestration of carbon via photosynthesis.

Further studies of the technical feasibility and sustainability (economic, environmental, and social) are needed to assess if the potential advantages of this system can be implemented in practice at full scale. An important issue is the scale-up of the system, which should be carried out taking advantage of the increased scale without raising further technical difficulties for the photosynthetic activity of the microalgae consortia.

Although the yields obtained from anaerobic digestion have not been the most promising, developments in the application of microalgae-based systems for wastewater treatment must be leveraged to meet the ambitious decarbonization targets needed to mitigate human-caused global climate change.

**Author Contributions:** Conceptualization, L.B., M.R.T., R.B., S.R. and J.V.; methodology, L.B., M.R.T., R.B., S.R., E.G.M. and J.V.; validation, L.B., M.R.T., R.B., S.R., E.G.M. and J.V.; formal analysis, M.R.T., R.B., S.R., B.R., V.A., P.G., J.M. and P.R.C.; investigation, L.B., M.R.T., R.B., S.R., E.G.M. and J.V.; data curation, L.B., M.R.T., R.B., S.R., E.G.M. and J.V.; writing—original draft preparation, R.B., E.G.M. and S.R.; writing—review and editing, L.B., M.R.T., R.B., S.R., E.G.M. and J.V.; supervision, L.B., M.R.T., R.B., S.R., E.G.M. and J.V.; project administration, L.B. and M.R.T.; funding acquisition, L.B., M.R.T., R.B., S.R. and J.V. All authors have read and agreed to the published version of the manuscript.

**Funding:** This research was funded by the Foundation for Science and Technology (FCT) through UIDB/04326/2020, UIDP/04326/2020, UID/00350/2020, LA/P/0101/2020 and LA/P/0069/2020 grants, and the GreenTreat (PTDC/BTA-BTA/31567/2017). CRESC-Algarve and the European Regional Development Fund (ERDF) programs funded the ALGAVALOR (ALG-01-0247-FEDER-035234) project.

**Institutional Review Board Statement:** Not applicable.

**Informed Consent Statement:** Not applicable.

**Data Availability Statement:** Not applicable.

**Acknowledgments:** The authors would like to thank Águas do Algarve, S.A., for their help and availability to receive the pilot installation of the GreenDune photobioreactors for the experiments.

**Conflicts of Interest:** The authors declare no conflict of interest.

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
