# Peer review of "Biogas Production from Microalgal Biomass Produced in the Tertiary Treatment of Urban Wastewater: Assessment of Seasonal Variations"

_energies, doi:10.3390/en15155713_

Round 1

Reviewer 1 Report

The paper deals with the anaerobic digestion of microalgae grown on wastewater as a tertiary treatment. The authors studied biogas production from different microalgal biomass (produced in different seasons during tertiary treatment) and pretreatment in order to increase biogas production from this biomass.

The topic of the paper has been under research for a long time, so several papers similar to this one can be found in the scientific literature. In any case, the paper adds more information regarding biogas production from microalgae grown on wastewater and composition variations during seasons under a pilot scale.

Below authors can find some specific comments:

 Section 2.1.

#1 Did the authors study and determine microalgae consortia species? This is a crucial factor to understand the results and draw appropriate conclusions.

 Section 2.2.

#2 Did the authors follow any standards for the execution of the BMP assays? Please specify if so.

There are some very important parameters that must be taken into account when executing BMP assays, such as substrate to inoculum ratio (in VS). So, if no standard was followed for the execution of the BMP assay, please specify these types of procedures in more detail.

 #3 Please, specify that you did use blank reactors to measure endogenous biogas production of the inoculum.

 Section 2.3.6.

#4 Did you check the necessary conditions for ANOVA analysis? Normally, in BMP assays with only 3 replicates (normal size) some of the conditions that populations should comply with are difficult to get:

-          Normal distribution

-          Equal variances

This could make an ANOVA analysis an inappropriate way to determine significant differences.

 Section 3.2.

#5 Please, check lines 217-218. In my opinion, the last sentence is not expressed properly. Biogas production from microalgae biomass cannot be defined so categorically since it depends on different aspects.

 #6 Lines 235-238.

As I understand, the authors are evaluating biogas and methane production in terms of VS. Therefore, this reasoning may not be appropriate or should be explained in more detail. This point is probably more related to organic matter composition and the type of microalgal biomass.

Moreover, the presence of predatory microorganisms, which are supposed to be harvested with microalgal biomass (if not please specify) could have even a positive effect if the biodegradability of the microalgal biomass is low due to the presence of cell walls that impede anaerobic biodegradation of complete biomass.

In my opinion, this paragraph should be improved.

 #7 Lines 260-262

Pretreatment did not show a uniform effect on methane production from microalgal biomass. Maybe, microalgae consortia are different during the different seasons (it is very probable) and that's why the authors did not find a uniform effect of the pretreatment. There is no universal pretreatment that is equally effective in disrupting the cell walls of different microalgal species. And very important, conditions of the pretreatment may be effective for some consortia but not for others. In my opinion, the results do not support this conclusion and the discussion should be more elaborated.

 #8 Line 278

Please, correct biomass for biogas

 #9 Lines 278-281

In my opinion, this sentence looks a bit contradictory, this should be explained better.

Author Response

The paper deals with the anaerobic digestion of microalgae grown on wastewater as a tertiary treatment. The authors studied biogas production from different microalgal biomass (produced in different seasons during tertiary treatment) and pretreatment in order to increase biogas production from this biomass.

The topic of the paper has been under research for a long time, so several papers similar to this one can be found in the scientific literature. In any case, the paper adds more information regarding biogas production from microalgae grown on wastewater and composition variations during seasons under a pilot scale.

We thank the reviewer for his/her valuable comments.

Below authors can find some specific comments:

Section 2.1.

Did the authors study and determine microalgae consortia species? This is a crucial factor to understand the results and draw appropriate conclusions.

Answer: The authors did indeed make a study concerning the composition of the consortia. However, this information will be published elsewhere, because of the enormous amount of data that was gathered across the seasons and the different HRTs. We can say already the dominant microalgal species were freshwater chlorophytes with lower biodiversity in the summer compared to those in the spring and autumn / fall. Interestingly, we found animal and microalgal parasites and other common predators of microalgae (ciliates and amoebae either from Amoebozoa or other evolutionary lines with amoeboid life stages).

Section 2.2.

#2 Did the authors follow any standards for the execution of the BMP assays? Please specify if so.

There are some very important parameters that must be taken into account when executing BMP assays, such as substrate to inoculum ratio (in VS). So, if no standard was followed for the execution of the BMP assay, please specify these types of procedures in more detail.

#3 Please, specify that you did use blank reactors to measure endogenous biogas production of the inoculum.

Answer to #2 and #3: Since the procedure used was not standard, some more experimental details have been included in section 2.2, including the specification of the use of inoculum blanks, the specification of the substrate to inoculum ratio (in SV) and clarification of the calculations of biogas production.

Section 2.3.6.

#4 Did you check the necessary conditions for ANOVA analysis? Normally, in BMP assays with only 3 replicates (normal size) some of the conditions that populations should comply with are difficult to get:

- Normal distribution

- Equal variances

This could make an ANOVA analysis an inappropriate way to determine significant differences.

Answer to #4: We have assumed that analysis of variance would be an appropriate method to gauge the statistical relevance of the differences observed. There is no evidence on the collected experimental results that counters this assumption.

 Section 3.2.

#5 Please, check lines 217-218. In my opinion, the last sentence is not expressed properly. Biogas production from microalgae biomass cannot be defined so categorically since it depends on different aspects.

Answer to #5: Following the reviewer’s recommendation, the sentence has been changed, reporting a range of reported results and some of the reasons for the variability observed.

#6 Lines 235-238.

As I understand, the authors are evaluating biogas and methane production in terms of VS. Therefore, this reasoning may not be appropriate or should be explained in more detail. This point is probably more related to organic matter composition and the type of microalgal biomass.

Moreover, the presence of predatory microorganisms, which are supposed to be harvested with microalgal biomass (if not please specify) could have even a positive effect if the biodegradability of the microalgal biomass is low due to the presence of cell walls that impede anaerobic biodegradation of complete biomass.

In my opinion, this paragraph should be improved.

Answer to #6: VS is and indirect measure of organic matter content and does not give information about its biodegradability under anaerobic digestion conditions. In the consortia grown, predation and parasitism are expected to consume a large proportion of the readily biodegradable OM of microalgae while still in the photobioreactors, leaving a higher proportion of recalcitrant material for anaerobic digestion. It is true that the predatory and parasitic organisms attack the cell walls of the microalgae, but this is a rate-limiting step on their actions inside the photobioreactors, so it is expected that the released biodegradable organics are quickly consumed. The amount of predatory/parasitic biomass grown is much lower than the microalgae biomass consumed by these processes. The text has been slightly modified to further clarify this point.

#7 Lines 260-262

Pretreatment did not show a uniform effect on methane production from microalgal biomass. Maybe, microalgae consortia are different during the different seasons (it is very probable) and that's why the authors did not find a uniform effect of the pretreatment. There is no universal pretreatment that is equally effective in disrupting the cell walls of different microalgal species. And very important, conditions of the pretreatment may be effective for some consortia but not for others. In my opinion, the results do not support this conclusion and the discussion should be more elaborated.

Answer to #7: Even though table 1 apparently shows that in some samples the methane production decreases (e.g., Autumn) and in others it increases (e.g., Winter24), statistical analysis shows these differences are not significant. Therefore, it is not correct for the reviewer to mention non-uniform effect of the pre-treatment. We agree that there is no universal pre-treatment usable with any microalgae consortia, however, we have shown before that thermoacidic pre-treatment is effective in the mobilization of organic matter content for ethanol fermentation in a wide range of microalgae species (see doi.org/10.1016/ j.algal.2016.12.021 or doi.org/10.1016/j.algal.2021.102329). We have also observed a large increase in reducing sugar content of all the biomass samples after thermoacidic pre-treatment (mention to this has been added on this manuscript). The observation that in the present case the methane yield does not increase leads us to the conclusion that the release of organic matter content from the cells by the inoculum is already efficient during anaerobic digestion. The text has been slightly changed to clarify this point. Table 1 has been simplified, omitting the statistical analysis of the differences between hydrolysed samples, because this point has no importance for our reasoning.

#8 Line 278

Please, correct biomass for biogas

Answer to #8: The correction was done, it was a indeed a writing glitch.

#9 Lines 278-281

 In my opinion, this sentence looks a bit contradictory, this should be explained better.

Answer to #9: The sentence was slightly modified to make it clearer. See also answer to comment #7.

Reviewer 2 Report

The paper was revised according to the journal rules. The paper was focused on the biogas production yield using microalgal biomass.

Few revisions are required and they are reported below:

- please add a nomenclature list with all acronyms, parameteres and unit of measure

- please revise the materials and methods section checking that all details for the instruments used are added to the manuscript

-

- in figure 2 the uncertainity bars were added, please add these considerations also in figure 1

- being an experimental study please add an unvertainity analysis section to describe better the results achieved

- add references in "Depending on the media...biofuels"

- add references in "The use of ... production"

- in addition to the biogas production, microalgae could be used also, in a biogas site, for the gas cleaning for the energy generations, add references such as (10.1007/s12649-019-00931-3, 10.1016/j.scitotenv.2016.06.1, 10.1016/j.biortech.2014.10.022, 10.1016/j.jcou.2017.01.014)

- future developments and technical feasibility should be considere in the conclusion section

Author Response

The paper was revised according to the journal rules. The paper was focused on the biogas production yield using microalgal biomass.

Few revisions are required and they are reported below:

- please add a nomenclature list with all acronyms, parameteres and unit of measure

Answer: A nomenclature list with all acronyms, parameters and units of measure is not required by journal rules; also, the number of acronyms and measurement units used in this manuscript is not unusually large, hence, the authors deemed that such list as suggested by the reviewer was not necessary. However, special care has been taken in this revised version to define all acronyms and symbols the first time they are used.

- please revise the materials and methods section checking that all details for the instruments used are added to the manuscript

Answer: All the details requested have been included.

- in figure 2 the uncertainity bars were added, please add these considerations also in figure 1

Answer: The type of graph used in the figure does not allow for error bars to be included. We opted for representing the data on a table instead (now Table 1).

 - being an experimental study please add an unvertainity analysis section to describe better the results achieved

Answer: As clearly stated in the materials and methods section all the experiments of anaerobic digestion were carried out in triplicate, as were the chemical analyses of the biomass and recovered biogas. Analysis of variance (ANOVA) was carried out to assess the significance of the differences observed on the average values of the experimental results. These are standard procedures to cope with the uncertainty of experimental work. In the authors opinion, they are adequate for the present study, and no special further section describing uncertainty analysis is needed.

 - add references in "Depending on the media...biofuels"

Answer: A reference was added as suggested.

 - add references in “The use of … production”

Answer: A reference was added as suggested.

 - in addition to the biogas production, microalgae could be used also, in a biogas site, for the gas cleaning for the energy generations, add references such as (10.1007/s12649-019-00931-3, 10.1016/j.scitotenv.2016.06.1, 10.1016/j.biortech.2014.10.022, 10.1016/j.jcou.2017.01.014)

Answer: A mention to these possibilities, and the respective references, have been included. We thank the reviewer for the valuable contribution.

 - future developments and technical feasibility should be considere in the conclusion section

Answer: As suggested by the reviewer, considerations about these aspects have been included in the conclusion section.

Reviewer 3 Report

The main issue of the reviewed manuscript was assessment of the microalgal biomass, derived from the third stage of the wastewater treatment system, in terms of the biogas production via anaerobic digestion. The topic is quite popular nowadays and SCOPUS search with phrases microalgae AND biogas gave 383 outputs. The reviewed article is a case study that should attract the attention of the scientific community, however before the acceptance several critical issues have to be addressed.

1)      Title is too long, there is definitely potential to make it shorter and expose crucial aspects

2)      Introduction - this section in general provides basic insights into the issue of the article, however better flow (conncetions between sections) of the presented ideas is required, as well wider explanation of the novelty of the presented experimental work is required. In the introduction section Authors emphasized that most published work has been conducted with monocultures of algae, while there are also plenty of papers related to the mixed cultures. In my opinion such explanation is not sufficient to emphasize article novelty, please expand.

3)      Materials & methods, in further parts of the article, the Authors present variability of the biomass composition during the year, but in my opinion the averaged composition of the influent wastewater or nutrient loads passing the reactor should be presented to better understand this variability. We got only information about HRT, but many information still missing. In addition, we are missing many details in the description of the sampling protocol and experiment design.

a.       What was a size of the photobioreactor and settler? Was it full, pilot, laboratory scale?

b.       What was averaged biomass concentration in the reactor as well over excessive biomass production rate? This issue also should be presented in more detailed way in the Results/Discussion (R&D) section point 3.3.

c.       What volume of the biomass sample was collected for further analysis?

d.       Please specify in the M&M section how the sampling was performed over the experimental periods. How many samples were collected? If for instance, summer sample was as single biomass sample digested in tree replications or more samples from the particular period were collected?

4)      In the R&D section 3.1. Authors pointed that biomass composition was affected by variations of the composition of the influent WW as well as the temperature and solar radiation. What about changes in the composition of the microalgae species? Maybe, some species prevail during particular season or they change their physiological properties in response to the variable conditions. Definitely, microbial studies missing in the reviewed article, therefore, please elaborate this issue with available literature data.

5)      Presented results showed that pretreatment did not increase the biogas production. Are there any other solutions to solve this issue? What about co-digestion or alternate inoculum?

6)      Lines 272-281 –present the most valuable output of the presented studies, therefore should be moved to the conclusion section.

Author Response

The main issue of the reviewed manuscript was assessment of the microalgal biomass, derived from the third stage of the wastewater treatment system, in terms of the biogas production via anaerobic digestion. The topic is quite popular nowadays and SCOPUS search with phrases microalgae AND biogas gave 383 outputs. The reviewed article is a case study that should attract the attention of the scientific community, however before the acceptance several critical issues have to be addressed.

We thank the reviewer for his/her valuable comments.

Title is too long, there is definitely potential to make it shorter and expose crucial aspects

Answer: We have proposed a new, shorter title.

Introduction - this section in general provides basic insights into the issue of the article, however better flow (conncetions between sections) of the presented ideas is required, as well wider explanation of the novelty of the presented experimental work is required. In the introduction section Authors emphasized that most published work has been conducted with monocultures of algae, while there are also plenty of papers related to the mixed cultures. In my opinion such explanation is not sufficient to emphasize article novelty, please expand.

Answer: The introduction has been modified by i) stressing out the advantages of using microalgae for tertiary wastewater treatment; ii) adding other examples of microalgae usefulness in biogas sites; iii) reducing the relative importance of the use of pre-treatments, which in this study were employed only to assess the biodegradability of the biomass; and iv) stressing out that the cultivation system consists of novel photobioreactors installed inline on the WWTP fed with real secondary wastewater.

Materials & methods, in further parts of the article, the Authors present variability of the biomass composition during the year, but in my opinion the averaged composition of the influent wastewater or nutrient loads passing the reactor should be presented to better understand this variability. We got only information about HRT, but many information still missing. In addition, we are missing many details in the description of the sampling protocol and experiment design.

    1. What was a size of the photobioreactor and settler? Was it full, pilot, laboratory scale?
    2. What was averaged biomass concentration in the reactor as well over excessive biomass production rate? This issue also should be presented in more detailed way in the Results/Discussion (R&D) section point 3.3.
    3. What volume of the biomass sample was collected for further analysis?
    4. Please specify in the M&M section how the sampling was performed over the experimental periods. How many samples were collected? If for instance, summer sample was as single biomass sample digested in tree replications or more samples from the particular period were collected?

Answer: The microalgal biomass was produced in pilot 480L GreenDune photobioreactors installed inline at the wastewater treatment plant of Quinta do Lago, Portugal. Further details regarding the design of the PBRs as well as a figure were added. Details regarding the performance of the PBRs on the tertiary treatment of the water are the object of another publication, currently submitted to a different journal. Nonetheless, the range of total nitrogen, nitrates, ammonium, and total phosphorus concentrations as well as the COD of the inflow water are reported in section 2.1.

In the R&D section 3.1. Authors pointed that biomass composition was affected by variations of the composition of the influent WW as well as the temperature and solar radiation. What about changes in the composition of the microalgae species? Maybe, some species prevail during particular season or they change their physiological properties in response to the variable conditions. Definitely, microbial studies missing in the reviewed article, therefore, please elaborate this issue with available literature data.

Answer: Season variation was indeed observed, in particular a very significant decrease in biodiversity of the microalgal consortia in the photobioreactors was observed in the summer compared to spring and autumn / fall. However, because the data was wide enough and interesting enough for another paper fully dedicated to the consortia composition, we will publish this information elsewhere, and of course we will integrate the information gathered in this and other papers we have published and will publish about this very subject. Another thing we observed is the presence of known microalgae predators and parasites, which makes these photobioreactors a very interesting case for studying the interaction between microalgae and these latter species when grown in wastewater.

Presented results showed that pretreatment did not increase the biogas production. Are there any other solutions to solve this issue? What about co-digestion or alternate inoculum?

Answer: The issue is not how to improve the biogas yield from this biomass. The pre-treatment studies point to the conclusion that this is very difficult to achieve, because the intrinsic biodegradability of the samples seems to be lower than that of optimally grown monocultures. The abstract has been modified to stress less on the failure of the pretreatment but rather point to the conclusions allowed by the pretreatment results.

Lines 272-281 –present the most valuable output of the presented studies, therefore should be moved to the conclusion section.

Answer: The suggested change was implemented.

Round 2

Reviewer 3 Report

The authors addressed all my comments and modified manuscript in satisfactory manner